# On the Differential and the Integral Value of Information

**DOI:** 10.3390/e27010043

**Published:** 2025-01-07

**Authors:** Raphael D. Levine

**Affiliations:** 1Institute of Chemistry, The Hebrew University of Jerusalem, Jerusalem 91904, Israel; raphy@mail.huji.ac.il; 2Department of Molecular and Medical Pharmacology, David Geffen School of Medicine, University of California, Los Angeles, CA 90095, USA; 3Department of Chemistry and Biochemistry, University of California, Los Angeles, CA 90095, USA

**Keywords:** mutual information, Lagrange multiplier, constraints on a probability distribution, cross correlation of constraints

## Abstract

A quantitative expression for the value of information within the framework of information theory and of the maximal entropy formulation is discussed. We examine both a local, differential measure and an integral, global measure for the value of the change in information when additional input is provided. The differential measure is a potential and as such carries a physical dimension. The integral value has the dimension of information. The differential measure can be used, for example, to discuss how the value of information changes with time or with other parameters of the problem.

## 1. Introduction

The fathers of what we now call information theory were quite clear. Information is to be defined in a manner that is objective. Information is provided by an answer to a question. The value of the answer for a particular person is not part of the theory. To represent the uncertainty before the answer is given, one considers a random variable *X*. Take it that there are *n* different possible answers *n* ≥ 2. And that the random variable *X* can assume *n* different values $\left\{x_{i}\right\},\;i=1,2,\ldots n,$ each one being a possible answer with a probability *p_i_*. The uncertainty about the answer (≡ entropy, *H*(*X*), of the random variable *X*) is reduced when additional information is provided. This reduction is the meaning of the term *value* as used in this paper. Our quantitative examination of the value of information is most influenced by the erudite discourse of this topic by Dunn and Golan [1]. There is, of course, the general result, introduced by Shannon [2], of the reduction in the uncertainty about the random variable *X* given another random variable Y. This concept is central to the celebrated channel capacity theorem of Shannon. In that context, the noisy channel that determines the conditional distribution of the output *Y* given an input *X* is fixed. It is the probability distribution of the input that one can vary. This central development of Shannon is extensively discussed in texts on information theory such as Ash [3] or Cover and Thomas [4] and many others. The general expression of Shannon for the information provided by *Y* about *X*
(1)$$I\left(X;Y\right)=H\left(X\right)-H\left(X|Y\right)$$
is not limited to information transmission by a channel. *H*(*X*) is the uncertainty (≡ entropy) about the random variable *X* while *H*(*X*|*Y*) is the remaining uncertainty about *X* when *Y* is given.

In this note, I assume that the additional information on *X* is provided by one or more additional expectation values over the distribution of *X*. Say that we are given one additional expectation value *F_l_* of an observable that takes the value $f_{l}\left(x_{i}\right)$ when the answer is $x_{i}$,
(2)$$F_{l}=f_{l}=\sum_{i=1}^{n}f_{l}\left(x_{i}\right)p_{i}$$

Before we are given this additional information, the random variable *X* is characterized by *m* expectation values, 1≤m<n, that we denote as $F_{1},F_{2},\ldots ,F_{m}$. We take it that the different vectors of *n* components $f_{1},f_{2},\ldots ,f_{m}$ are linearly independent. Otherwise, some of them are redundant and do not provide additional information about the distribution of *X*. By a similar reasoning, the vector $f_{l}$, whose expectation value Equation (2) is used to specify the additional information, needs to be linearly independent of the *m* vectors $f_{1}$ to $f_{m}$. Of course, we need to know that the *m* expectation values $F_{1}$ to $F_{m}$ are compatible, meaning that there is one or more distribution for which these are possible expectation values. This is also for the additional information, meaning the expectation value *F_l_*. Our program, as implemented in the Results section, is to determine the value of the information on *X* provided by the additional information. I will discuss two views of the result, a differential and an integral value. For either, I take it that the prior information about *X*, namely, the values of the *m* expectations $F_{1}$ to $F_{m}$, are kept constant. The differential form is a partial derivative, an infinitesimal change in the entropy of *X* upon an infinitesimal change in the value of the new information, the value of *F_l_*, when all the other *m* expectation values are unchanged. The integral form is a change in the entropy due to a finite change in the value of *F_l_* again, at constant value of the prior information. The distinction between a differential and an integral change is of course familiar in many other contexts of the exact sciences perhaps most notably so in quantum chemistry where the expression for differential change in the energy is known as the Hellmann–Feynman theorem [5]. One can also apply the theorem for changes in the dynamics, e.g., [6].

The presentation is organized as follows. Section 2, Methods, provides essential results of the maximum entropy formalism [7,8,9,10,11]. We use the *m* expectation values $F_{1}$ to $F_{m}$ as constraints on the entropy of the distribution of *X*. Among all distributions over the *n* different values $\left\{x_{i}\right\},\;i=1,2,\ldots n$ that are consistent with the *m* expectation values $F_{1}$ to $F_{m}$, *m* < *n*, we determine the (unique) distribution whose entropy is maximal. We take it that there is a feasible solution, meaning that the values of the *m* expectation values are such that there is one or more distribution with all probabilities non zero that reproduces these values. The expectation value of *F_l_* is not imposed as a constraint on the distribution. Technically, the constraints are imposed by the Lagrange method of undetermined multipliers. The numerical value of the *m* Lagrange thus far undetermined multipliers $\lambda_{1}$ to $\lambda_{m}$ is determined at the last stage by the condition that the distribution reproduces the *m* expectation values. For the observable that takes the value $f_{l}\left(x_{i}\right)$ on the *i*’th outcome whose probability is $p_{i}^{0}$, its expectation value as determined by the maximal entropy procedure subject to the *m* constraints is $F_{l}^{0}=\sum_{i=1}^{n}f_{l}\left(x_{i}\right)p_{i}^{0}$. Our purpose, as already introduced above, and as discussed in technical detail in Section 3, Results, is to determine the amount of information provided when the expectation value $F_{l}$ is changed from $F_{l}^{0}$. Section 4, The Value of Information as a Potential, provides motivation for the possible implications of the expression of the value of information, specifically for when the new information changes with time, a case of particular relevance for systems not in equilibrium. Also examined is a more formal issue when the additional information depends on a parameter.

## 2. Methods

To have a compact derivation, we take the variable *X* to be such that its distribution is uniform when entropy is maximal and no additional information is available. Shannon and then many others have shown that if we take as an axiom that when the distribution is uniform the entropy is maximal, then with a few additional axioms, the entropy of *X* is $H\left(X\right)=-\sum_{i=1}^{n}p_{i}\ln p_{i}$. The use of a natural logarithm just determines the units of entropy, nats in this case. In the physical sciences, it is usually the case that there is always some information, e.g., the conservation of energy, and so the distribution at maximal entropy is not uniform. The needed modification is well understood, and so we proceed with the most elementary case as above. As is emphasized early on, e.g., by Tolman [12], this is the case when the index *i* enumerates individual quantum states.

The distribution when the entropy is at a constrained maximum, i.e., at a maximum where the search for a maximum is subject to given *m* values $F_{1}$ to $F_{m}$, *m* < *n* is [7,8,9,10,13]
(3)$$p_{i}^{0}=\exp\left(-\sum_{k=1}^{m}\lambda_{k}f_{k}\left(x_{i}\right)\right)$$

The numerical values of the *m* Lagrange multipliers $\lambda_{k}$ are determined by the *m* implicit equations
(4)$$F_{j}=\sum_{i=1}^{n}f_{j}\left(x_{i}\right)p_{i}^{0}=\sum_{i=1}^{n}f_{j}\left(x_{i}\right)\;\exp\left(-\sum_{k=1}^{m}\lambda_{k}f_{k}\left(x_{i}\right)\right),\;j=1,2,\ldots ,m$$

The distribution needs to be inherently normalized, meaning that $\sum_{i=1}^{n}p_{i}^{0}=\langle 1\rangle =1$. So, either one of the observables is the identity, i.e., $f_{j}\left(x_{i}\right)=1\;\mathrm{for}\;\mathrm{all}\;i\;\mathrm{and}\;F_{j}=1$, or some linear combination of observables is the identity. Either way, the normalization is enforced so that $\partial 1/\partial F_{j}=0$. Since the values of the Lagrange multipliers are determined by the value of the *m* observables, it follows from $\partial \sum_{i=1}^{n}p_{i}^{0}/\partial F_{j}=0$ and Equation (3) that
(5)$$\sum_{k=1}^{m}(\partial \lambda_{k}/\partial F_{j})F_{k}=0$$

The entropy of the distribution of *X* is
(6)$$H\left(X\right)=-\sum_{i=1}^{n}p_{i}^{0}\ln p_{i}^{0}=\sum_{k=1}^{m}\lambda_{k}F_{k}$$

Taking the partial derivative of *H*(*X*) wrt $F_{j}$ and using the implication of the normalization, we have the basic identity for the value of information
(7)$$\lambda_{j}=\partial H\left(X\right)/\partial F_{j}$$

The value $\lambda_{j}$ is defined as a partial derivative when all the *m*-1 observables that are not $F_{j}$ have their value kept constant. The value carried dimension, those of $1/F_{j}$ or $\lambda_{j}F_{j}$, has the dimension of information, $\lambda_{j}F_{j}=F_{j}\;\partial H\left(X\right)/\partial F_{j}$. It follows from Equation (7) that the entropy of *X* is a homogeneous first-order function of the *m* observables that are used to constrain the distribution at its maximal possible entropy
(8)$$H\left(X\right)=\sum_{k=1}^{m}F_{k}\;(\partial H\left(X\right)/\partial F_{k})$$

Equation (8) generalizes a known result in thermodynamics [10,14,15,16]. The more general result, Equation (6), is that the entropy is the weighted sum of the values of the observables that define the state. It is a macroscale analog of the basic microscale definition of the entropy as the weighted sum of the surprisals, $-\ln p_{i}$, of the different possible outcomes. We call the value, $\lambda_{j}$, a potential because as was just shown, it is the latent ability of the observable conjugate to $\lambda_{j}$ to change the entropy.

## 3. Results

So far, we have examined the value of information when the value of one of the observables that defines the state is changed. In Equation (7), it is the change in the value of observable j. Next, we consider the value of the information provided when an observable *F_l_* is changed, and that observable was not used previously to characterize the state. As already noted, such an observable was not used because the distribution we assigned at a given value of *m* other observables already correctly produces its current value $F_{l}^{0}=\sum_{i=1}^{n}f_{l}\left(x_{i}\right)p_{i}^{0}$. This is a rather common situation. A well-known example is the Boltzmann distribution at thermal equilibrium. It is normalized and subject to the given (expectation) value of the energy of the different quantum states *i*. Given the mean, the distribution of quantum states correctly predicts the variance of the energy, which is the specific heat. Indeed, that was a very early success of the then new quantum theory.

Given the observable $F_{l}^{0}$, the value of making an infinitesimal change in its expectation is zero. This is because the distribution of *X* is at maximal entropy subject to the given expectations of *m* constraints. These values are to be held constant when we make an infinitesimal change in $F_{l}^{0}$. But linear variations about a stationary point of a function do not change it. A finite change, from $F_{l}^{0}$ to *F_l_*, does lead to a new distribution of maximal entropy, and Equation (3) is replaced by
(9)$$p_{i}^{\prime }=\exp\left(-\sum_{k=1}^{m}\lambda_{k}^{\prime }f_{k}\left(x_{i}\right)-\lambda_{l}^{\prime }f_{l}\left(x_{i}\right)\right)$$

We use primes to denote a distribution of maximal entropy subject to all the previous *m* constraints. $F_{1}$ to $F_{m}$ and to the new constraint $F_{l}$, where in Equation (9) *k* runs from 1 to *m*, excluding *l.* The value of the *m* expectation values remains unchanged when we go from the distribution $p_{l}^{o}$, Equation (3), to the distribution $p_{i}^{\prime }$, Equation (9), but the *m* Lagrange multipliers can change, and that is why they have a superscript prime,
(10)$$\partial \lambda_{j}^{\prime }/\partial F_{l}=\partial^{2}H\left(X\right)/\partial F_{j}\partial F_{l}$$

To characterize the value of the new information, we use the Shannon fundamental result, Equation (1) plus the inequality $\sum_{i=1}^{n}p_{i}^{\prime }\ln\left(p_{i}^{\prime }/p_{i}^{o}\right)\geq 0$, where equality is iff $p_{i}^{\prime }=p_{i}^{o}$ for all *i*. Using the explicit Expressions (3) and (9) and noting that by construction, the *m* observables have the same expectation values for $p_{i}^{o}\;\mathrm{and}\;p_{i}^{\prime }$, we have
(11)$$\sum_{i=1}^{n}p_{i}^{\prime }\ln\left(p_{i}^{\prime }/p_{i}^{o}\right)=-\sum_{i=1}^{n}p_{i}^{o}\ln p_{i}^{o}-\left(-\sum_{i=1}^{n}p_{i}^{\prime }\ln p_{i}^{\prime }\right)=H\left(X\right)-H\left(X|Y\right)=I\left(X;Y\right)\geq 0$$

$H\left(X\right)$ is the entropy of the random variable *X* before the new information is provided, while the entropy $H\left(X|Y\right)=-\sum_{i=1}^{n}p_{i}^{\prime }\ln p_{i}^{\prime }$ is the entropy of *X* after the additional information *Y*, that is here the value of $F_{l},$ is provided. As is to be expected on general grounds, the value of the new information is positive unless $p_{i}^{\prime }=p_{i}^{o}$, meaning that the new information is not really informative as the distribution of *X* is unchanged.

Explicitly, the finite value of the new information is, using the expressions (3) and (9),
(12)$$I\left(X;Y\right)=\sum_{i=1}^{n}p_{i}^{\prime }\ln\left(p_{i}^{\prime }/p_{i}^{o}\right)=-\sum_{k=1}^{m}(\lambda_{k}^{\prime }-\lambda_{k})F_{k}-\lambda_{l}^{\prime }(F_{l}-F_{l}^{o})$$

The differential change in the Lagrange multipliers due to the change in the value $F_{l}$ is provided by Equation (10). It is a vector with *m* component indexed by *j*. It is a vector orthogonal to the constraints, as shown in Equation (5).

Any information that is provided adiabatically has no value. In general, this follows from the Shannon definition, Equation (1), of the information provided by *Y* about *X*. $I\left(X;Y\right)=0$ when the distribution of *X* is unchanged when *Y* is given, $H\left(X|Y\right)=H\left(X\right)$. On the microscale, that is in terms of the individual outcomes, this is $p\left(x_{i}|y_{j}\right)=p\left(x_{i}\right)$ for all *i* and *j*. This shows that our use of the term adiabatic follows the conventional use in thermodynamics and mechanics: the elementary probabilities of the different outcomes do not change upon a change in the macro scale. But how can that be? We take a clue from a differential form of the first law of thermodynamics to write for an infinitesimal change in an observable
(13)$$\delta F_{k}=\sum_{i=1}^{n}f_{k}\left(x_{i}\right)\delta p_{i}+\sum_{i=1}^{n}p_{i}\delta f_{k}\left(x_{i}\right)$$

It follows that if the addition of information is such that $\delta F_{k}=\sum_{i=1}^{n}p_{i}\delta f_{k}\left(x_{i}\right)\equiv \langle \delta f_{k}\rangle$, then the probabilities of the elementary events are unchanged and there is no value to the information provided. In the case of the first law of thermodynamics *F* is the energy, the $f\left(x_{i}\right)\prime s$ are the energies of the individual states. Then, 〈δf〉 is the work performed on or by the system, while δf−〈δf〉  is the heat transfer, which is zero in an adiabatic change. Performing pure work on or by a system does not change the value of the information of its state. The transfer of heat does, as was early and clearly noted by Clausius [17].

## 4. The Value of Information as a Potential

The value of information, the Lagrange multiplier that is conjugate to the observable that is changes, is a potential. We here discuss it as a potential, see also [13,18], and then specifically discuss how the value changes with time in the special but important case of Hamiltonian dynamics. Given that the state of the system is of maximal entropy, the set of *m* values of the constraints and the set of *m* Lagrange multipliers that is conjugate are each an equally correct and useful characterization of the state. As discussed by Callen [14] (the first edition is much better in this particular respect), there is a Legendre transform relating the use of the two sets of variables. As clearly discussed by Callen, one can also usefully introduce intermediate characterizations, using some observables and the other variables being the rest of the Lagrange multipliers. The practical advantages of using certain thermodynamic Lagrange multipliers such as temperature or pressure are well recognized. In chemical problems, there are the chemical potentials of the different species. In general, it is the *intensive* character of the Lagrange multipliers that often makes them more convenient. We use intensive in its canonical thermodynamic terminology: intensive meaning independent of the actual amount as opposed to the extensive character of the mean value of the observables that will double when we double the number of systems.

The change in the value in a process is often a useful measure. Already noted is that the value is unchanged in an adiabatic process. How does the value change in time? The problem in making a definitive answer is that we only have mechanics as an agreed upon dynamical theory of change, and in mechanics, both classical and quantum mechanical, the processes are reversible. There is no dissipation. Realistically, we all recognize that there is dissipation. Still, what can one say about the rate of change in value in mechanics. I will use quantum mechanics and, as a practical point, I assume a Hilbert space of finite dimensions *n*. So, operators and in particular the density operator are represented by *n* by *n* matrices. It will be useful below to use the *n*^2^ operators $E_{ij}$ defined as a matrix where all elements are zero except the element in position *i*,*j* that is unity. Any *n* by *n* matrix can be expressed as a linear combination of these matrices. As a side comment, these operators close a Lie algebra, $\left[E_{ij},E_{kl}\right]=E_{il}\delta_{jk}-E_{kj}\delta_{li}$, where the square bracket is the commutator and the delta symbol is the usual Kronecker delta.

The most general form of an initial state density matrix is
(14)$$\rho \left(0\right)=\exp\left(-\sum_{i,j=1}^{n}\Lambda_{ij}\left(0\right)E_{ij}\right)$$
where the multipliers $\Lambda_{ii}$ are real, while, since $E_{ji}=E_{ij}^{†}$ and the density needs to be Hermitian, the off diagonal coefficients must satisfy $\Lambda_{ji}=\Lambda_{ij}^{*}$. The observables $F_{k}$ of Equation (3) and following can be expressed as linear combinations of the ${E_{ij}}^{\prime }s$.

In quantum dynamics, the density matrix of an isolated system evolves in time under the action of a unitary evolution operator $U\left(t\right)$ with the initial value $U\left(0\right)=I$. Since *U* is unitary and using a superscript dagger to denote a Hermitian conjugate, we can write
(15)$$\rho \left(t\right)=U\left(t\right)\exp\left(-\sum_{i,j=1}^{n}\Lambda_{ij}\left(0\right)E_{ij}\right)U^{†}\left(t\right)=\exp\left(-\sum_{i,j=1}^{n}\Lambda_{ij}\left(0\right)\left(U\left(t\right)E_{i,j}U^{†}\left(t\right)\right)\right)$$

An initial density matrix of maximal entropy that is propagated in time remains a density matrix of maximal entropy with a different, time-dependent set of constraints [19]. These constraints can be shown to be time-dependent constants of the motion (see, e.g., [20]).

In a finite, *n*, dimensional Hilbert space, the time-dependent operators are *n* by *n* Hermitian matrices. So, they can all be written as linear combinations of the time-independent matrices $E_{ij}$ with time-dependent coefficients
(16)$${É}_{ij}\left(t\right)\equiv U\left(t\right)E_{i,j}U^{†}\left(t\right)=\sum_{kl,=1}^{n}e_{ij,kl}\left(t\right)E_{kl}$$

Thereby, the density at time *t*, Equation (15), can be written as
(17)$$\rho \left(t\right)=\exp\left(-\sum_{i,j=1}^{n}\Lambda_{ij}\left(0\right){É}_{ij}\left(t\right)\right)=\exp\left(-\sum_{k,l=1}^{n}\Lambda_{kl}\left(t\right)E_{kl}\right)$$
where the multipliers evolve as
(18)$$\Lambda_{kl}\left(t\right)=\sum_{i,j=1}^{n}\Lambda_{ij}\left(0\right)e_{ij,kl}\left(t\right)$$

The multipliers evolve contra-gradient to the observables. This is the most general equation of change for the values of the information provided by the basis observables $E_{kl}$. The one cardinal assumption is that the dynamics are unitary as used in Equation (14). It is a strong assumption because unitary implies reversible, $U\left(-t\right)=U^{†}\left(t\right)=U^{-1}\left(t\right)$. Even if $U\left(t\right)$ is not unitary, as long as the dynamics keep the system confined to the *n* dimensional Hilbert space, it is still possible to expand the surprisal matrix, $-\ln\rho \left(t\right)=\sum_{k,l=1}^{n}\Lambda_{kl}\left(t\right)E_{kl}\;$ in the complete basis of *n*^2^ observables, with time-dependent coefficients. Thereby, the $\Lambda_{kl}\left(t\right)\prime s$ remain their significance as the time-changing values of the elementary observables $E_{kl}$, but the system no longer evolves in a reversible manner.

## Data Availability

No new data were created or analyzed in this study. Data sharing is not applicable to this article.

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
