# Peer review of "On the Differential and the Integral Value of Information"

_entropy, 2025, doi:10.3390/e27010043_

Round 1

Reviewer 1 Report

Comments and Suggestions for Authors

See attached file.

Reviewer 2 Report

Comments and Suggestions for Authors

In this study, a quantitative expression for the value of information within the framework of information theory and of the maximal entropy formulation is explored. The author studied both a local, differential measure and an integral, global measure for the value of the change in information when additional input is provided. The author found that the differential measure is a potential and carries a physical dimension. The author also found that the integral value has the dimension of information. The differential measure can be used to discuss how the value of information changes with time or with other parameters of the problem. The paper is clearly written with significant results. I have a few questions before its publication. 

1. What is the relationship between the author's information measures and the other already existing information measures. What are the advantage of using the new measures compared to the existing measures?

Reviewer 3 Report

Comments and Suggestions for Authors

Referee's Report on Manuscript "On the Differential and Integral Value of Information" by Raphael D. Levine

Submission to: "Entropy"

The aforementioned paper presents a framework for analysis of the value of information when additional input is considered. Here, value of information represents the decrease in uncertainty (=entropy), and additional input is represented by statistical expectations of certain functions of the random variable.
This approach is useful when applying, e.g. maximum of entropy principle to various problems.
The paper is well-written and I recommend its acceptance in Entropy.

Minor comments and suggestions the author is free to ignore:

1. I enjoyed reading the manuscript written by such an authority in this field. Nevertheless, I would still use third person when referring to the author: end of line 34 "My quantitative" -> "Our quantitative"

2. I agree with equations (5) and onward when the critical point (p_1^0,...,p_n^0) is interior to the unit simplex. But what happens if the critical point is on the boundary? Essentially, this amount to a Lagrange multiplier enforcing p_i^0 >= 0. Would the KKT conditions require additional parameters (slack parameters?)? For instance, consider the case X in {1,2,3} and f1=E[X]=3. Then the probability vector p0 is (0,0,1). The partial derivative of p0 w.r.t. f1 does not exist (at least on one side).

3. In equation (9): does the author mean k runs from 1 to m, excluding ell ?

4. In Equation (12): should F_k be replaced with F_k^o ? (In particular, for k=ell ?)

Round 2

Reviewer 1 Report

Comments and Suggestions for Authors

Accept.